# Suicide Attempts among School-Attending Adolescents in Mongolia: Associated Factors and Gender Differences

**DOI:** 10.3390/ijerph19052991

**Published:** 2022-03-04

**Authors:** Javzan Badarch, Bayar Chuluunbaatar, Suvd Batbaatar, Edit Paulik

**Affiliations:** 1Department of Public Health, Albert Szent-Györgyi Medical School, University of Szeged, 6720 Szeged, Hungary; paulik.edit@med.u-szeged.hu; 2Medicine and Medical Devices Regulatory Agency, Ulaanbaatar 14210, Mongolia; bayar@mmra.gov.mn; 3Department of Environmental Health, National Center for Public Health, Ulaanbaatar 13381, Mongolia; suvd552001@gmail.com

**Keywords:** prevalence, suicide, attempt, adolescents, health survey, Mongolia

## Abstract

Attempting suicide is an important risk factor that can lead to suicide death. The aim of the current study was to examine the prevalence of suicide attempts and to identify the gender-specific predictors of suicide among adolescents in Mongolia. We analyzed data from the 2019 Mongolian Global School-Based Health Survey (GSHS) conducted nationwide among 13–18-year-old students. Univariable and multivariable analyses were performed to assess the correlates of suicide attempts. Overall, 32.1% of the adolescents reported to have had suicide attempts. Multivariable analysis showed a significant association in the total sample of suicide attempts with lack of close friends, anxiety, injury and violence, smoking and alcohol drinking, and sexual intercourse. Male suicide attempters were less likely to have close friends and more likely to have injuries, been physically attacked, been bullied, smoke, drink alcohol, and have had sexual intercourse. Within the female subgroup, anxiety, injury and violence, smoking and alcohol drinking significantly increased the odds of reporting suicide attempts. Increase of the student’s age by one year decreased the odds ratio of suicide attempts. Nearly one in three students had had a suicide attempt. Several factors, including mental distress, violence, and risky behaviors were found to be associated with suicide attempts. These can aid in designing intervention strategies for preventing suicidal behaviors among adolescents.

## 1. Introduction

Suicide and suicide-related behavior in young people have become serious and urgent global public health problems. More than 700,000 people in the world lose their life each year as a consequence of suicide [1]. In 2016, more than one in every 100 deaths (1.3%) was the result of committing suicide, and among individuals aged between 15 to 19 years, it was the third-leading cause of mortality. In addition, most of the world’s suicides occurred in low and middle-income countries (LMICs) (79%) [2], and in 2016, globally, more than 62,000 youths committed suicide [3].

Data on the prevalence of suicide attempts are available from various countries. Asia is the continent with the largest population size, and more than 60% of the world’s suicides occur in Asia, with China, India, and Japan being the most significant contributors to global suicide counts [4]. In Mongolia, mental health is second among the top five challenges that children are facing. The average suicide rate of Mongolian adolescents is five times as high as that of East Asia or the Pacific region. Moreover, in Mongolia, the suicide mortality among the young aged 10–14 years increased from 3.3% in 2003 to 11.4% in 2019 [5]. 

According to the statistical data, suicide rates among the young are alarming. For instance, the standardized suicide rates increased from 3.5 to 5.3 per 100,000 between 2001 and 2010 in South Korea [6], and from 9.1% to 11.3% between 2004 [7] and 2012 [8] in Taiwan, respectively. On average, 450 suicides occurred each year in Mongolia [9] and with 23.3 suicides per 100,000 inhabitants in 2016, the country ranked third in the world. A significant increase in suicide rates among adolescents (15–19 years) was found in Mongolia. In the 2010 Mongolian Global School-Based Health Survey (GSHS), the prevalence of suicide attempts was 8.7% [10]. Davaasambuu et al. found, in 2013, that about 10% of students had attempted suicide in the past 12 months, and students who lived in urban areas were more likely to have had a suicide attempt (11.8% vs. 8.6%) [11].

Suicide and suicide-related behaviors are multifactorial and complex. Many studies have shown that demographic variables, mental distress, violence, and risky behaviors, including substance use, are associated with increased risk of suicidal behavior in youth. Demographic factors connected with a suicide attempt may include being a female [12], older age [13], and urban locations [14]. Mental distress, including lack of close friends [15], anxiety-induced sleep disturbance [16], feeling lonely [17], and exposure to bullying/interpersonal violence [18] including suffering a physical attack [19], as well as a serious injury [20] have been found in a number of studies to be associated with adolescent suicide attempts. Risky behaviors associated with suicide attempts included substance use such as smoking [21] or alcohol drinking [22], as well as sexual intercourse [23]. 

Gender differences seem to play a crucial role in suicidal behavior of young people. Female adolescents are more prone to show internalizing disorders (e.g., anxiety) which may mediate the connection with suicidal behaviors [24], and females tend to have more suicide attempts than males [25]. In contrast, completed suicide was more frequent in males [26], which may be associated with a higher prevalence of externalizing disorders (e.g., substance-abuse disorder) [27].

Although findings of studies using the Mongolian GSHS 2010 and 2013 revealed that suicide attempts had increased significantly, no suicide-prevention program had been implemented in Mongolia in the previous decade. Considering this, the aim of the current study was to examine the prevalence of self-reported suicide attempts and to identify the gender specific predictors among school-attending adolescents in Mongolia using data from the GSHS 2019.

## 2. Materials and Methods

### 2.1. Participants and Procedures

The study involved data from the 2019 Mongolia GSHS. The Mongolian GSHS protocol was approved by Resolution No. 88 of the Ethical Committee of National Center for Public Health in November 2018. Students were asked to participate voluntarily in the survey, and a written informed consent was obtained from each student and parents/guardians. A two-stage cluster sample design was used to collect data to represent all students from 10 to 18 years of age in Mongolia. In the first stage of sampling, schools were selected with probability proportional to their reported enrolment size. In the second stage, classes in the selected schools were randomly chosen, and all students in the selected classes were eligible to participate. Altogether 4514 students participated in the 2019 Mongolian GSHS survey. Two types of questionnaires were applied, depending on the age of the target population (10–12-year-old or 13–18-year-old). The ten core GSHS questionnaire modules address the leading causes/risk factors of morbidity and mortality among children: tobacco, alcohol and other drug use; dietary behaviors; hygiene; mental health; physical activity; sexual behaviors that contribute to HIV infection, other sexually transmitted infections and unintended pregnancy; unintentional injuries and violence; and respondent demographics [28]. The question about attempted suicide was asked of the 13–18-year-old students. This question was answered by 2850 students, so in the present analysis, these students were considered as total.

### 2.2. Measures

With the exception of age, all variables were dichotomized as yes or no answers.

Suicide attempts: “During the past 12 months, how many times did you actually attempt suicide?” (Response options were from 1 = 0 times to 5 = 6 or more times; coded 1 = no and 2–5 = yes).

The independent variables were demographic factors, mental distress, injury and violence, and risky behaviors. 

#### 2.2.1. Demographic Factors

Gender: “What is your sex?” (Response options were 1 = male and 2 = female).

Age: “How old are you?” (Age in years).

#### 2.2.2. Mental Distress

Close friend: “How many close friends do you have?” (Response options were from 1 = 0 to 4 = 3 or more; coded 1 = no and 2–4 = yes).

Anxiety-induced sleep disturbance: “During the past 12 months, how often have you been so worried about something that you could not sleep at night?” (Response option were from 1 = never to 5 = always; coded 1–2 = no and 3–5 = yes).

Loneliness: “During the past 12 months, how often have you felt lonely?” (Response option were from 1 = never to 5 = always; coded 1–2 = no and 3–5 = yes).

#### 2.2.3. Injury and Violence

Being bullied: “During the past 30 days, on how many days have you been bullied?” (Response options were from 1 = 0 day to 7 = all 30 days; coded 1 = no and 2–7 = yes). (Description provided in the questionnaire: Bullying occurs when one or more students or someone else about your age teases, threatens, ignores, spreads rumors about, hits, shoves, or hurts another person over and over again).

Physically attacked: “During the past 12 months, how many times have you been physically attacked?” (Response options were from 1 = 0 time to 8 = 12 or more times; coded 1 = no and 2–8 = yes). (Description provided in the questionnaire: A physical attack occurs when one or more people hit or strike someone, or when one or more people hurt another person with a weapon (such as a stick, knife, or gun)).

Injury: “During the past 12 months, how many times have you been seriously injured?” (Response options were from 1 = 0 time to 8 = 12 or more times; coded 1 = no and 2–8 = yes). (Description provided in the questionnaire: An injury is serious when it makes you miss at least one full day of usual activities (such as, school, sports, or a job) or requires treatment by a doctor or nurse).

#### 2.2.4. Risky Behaviors

Cigarette smoking: “During the past 30 days, on how many days have you smoked cigarettes?” (Response options were from 1 = 0 day to 7 = all 30 days; coded 1 = no and 2–7 = yes).

Alcohol drinking: “During the past 30 days, on how many days have you had at least one drink containing alcohol?” (Response options were from 1 = 0 day to 7 = All 30 days; coded 1 = no and 2–7 = yes).

Sexual intercourse: “Have you ever had a sexual intercourse?” (Response options were 1 = yes and 2 = no).

### 2.3. Data Analysis

Data analysis was carried out with IBM SPSS (Statistical Package for the Social Sciences) version 27 (SPSS Inc., Chicago, IL, USA). Descriptive statistics including frequency, percentage, median, and interquartile range (IQR) were done to describe the study sample. Univariable logistic regression analyses were conducted to examine unadjusted associations between suicide attempts and independent variables. Multivariable forward stepwise logistic regression analysis was used to assess the independent contribution of demographic factors, mental distress, injury and violence, and risky behaviors to suicide attempts. The independent variables involved in the regression analysis were student age, place of residence, having close friends, anxiety, loneliness, being bullied, being physically attacked, injury, cigarette smoking, alcohol drinking, and sexual intercourse. Student age was considered as a continuous variable in the model. Odds ratio (OR) and 95% CI of OR were used to indicate the association between the suicide attempts and the selected list of independent variables. Statistical significance was defined at *p* < 0.05. Nagelkerke R^2^ values were used to evaluate the explanatory power of the models.

## 3. Results

A total of 2850 subjects were eligible for the analysis. The sex distribution showed female dominance (56% women and 44% men). The characteristics of the sample (total, male, and female) are shown in Table 1. The prevalence rate of attempted suicide was 32.1% in the total sample, 33.3% in males, and 31.3% in females. 

According to the univariable analysis (Table 2), male suicide attempters were less likely to have older age; more likely to be bullied, physically attacked, injured, smoke cigarettes, and drink alcohol. Female suicide attempters were less likely to have older age; more likely to live in urban location, have anxiety, feel lonely, be bullied, be physically attacked, be injured, smoke cigarettes, and drink alcohol. In the total sample, all factors showed a significant relationship with suicide attempts except sexual intercourse. Female students who had no close friends were less likely to have suicidal behavior than female students who had 1 or more close friends. The highest odds were found in connection with the risky behaviors, and with the injury and violence factors, especially among females.

The last step of the stepwise logistic regression models is shown in Table 3. Compared with the results of univariable analyses, the multiple models showed small differences in the predictors of attempted suicide in males and females. The living place and feeling lonely were not significant predictors in all models. Age remained significant in the case of females; each one year increase in age was associated with progressively fewer suicide attempts (AOR: 0.84). Anxiety and feeling lonely were not involved in the stepwise model in males, as was expected from the univariable results (*p* > 0.05), while in females, anxiety was a significant predictor (AOR: 2.02). In males, attempted suicide was more likely among those having sexual intercourse (AOR: 2.14). Altogether, male suicide attempters were less likely to have close friends, and more likely to having been bullied, physically attacked, injured, smoke cigarettes, and drink alcohol, and have had sexual intercourse. Within the female subgroup, lack of close friends, anxiety, being bullied, being physically attacked or injured, cigarette smoking, and alcohol drinking significantly increased the odds of reporting a suicide attempt.

## 4. Discussion

This was a cross-sectional study among school-attending adolescents based on the 2019 Mongolian GSHS. Mongolia belongs to the Western Pacific (WPR) region, which exhibited the highest overall prevalence of suicide attempts [29]. The prevalence of suicide attempts is a major concern in LMICs among young people aged 10–19 years. In this study, the prevalence of suicide attempts during the previous 12 months was 32.1% (33.3% for the males and 31.3% for the females), which was much higher than in previous studies based on the Mongolian GSHS of 2010 (8.7%) [10] and 2013 (10%) [11]. In addition, this result was also higher than the 12-month prevalence of suicide attempts in China (9.4%) [30], the Philippines (15.34%) [31], Vietnam (21.2%) [32], or Thailand (13.3%) [33], but lower than in the Solomon Islands (36.9%) [34].

In a previous study [13] based on the GSHS data of 82 countries from 2003 to 2015, it was found that the incidence of suicide attempts increases with age between 12 and 17 years. In contrast, our results showed that older age students (between 13 and 18 years of age) were less likely to have suicidal behavior.

This study found that having no close friends was a risk factor for suicide attempts among adolescents. It is well-known that having no close friends is associated with poor mental health including suicide attempts [35]. Conversely, school-going students who had support from their classmates were protected from experiencing suicide attempts [36].

Adolescents who had worry-induced sleep disturbance had higher odds of attempted suicide [37]. In this case, sleep disturbance is likely to be a symptom of anxiety. It has been reported that adolescents showing high levels all characteristics of anxiety disorder, including low distress tolerance and uncontrolled emotion, tend to have suicide attempts [38]. Anxiety in itself was found to be correlated with attempted suicide in the present study.

Findings from this study suggest that being bullied was strongly associated with suicide attempts. Being bullied is known to increase the risk of mental health problems, including poor motivational control, which may lead to increased risk of adolescent suicide attempts [39]. In order to prevent suicide in this population, it will also be crucial to improve anti-violence interventions.

The results of this study, namely that the social adversities of being physically attacked increased the odds of suicide attempts, were consistent with evidence from the GSHS in other South East Asian countries, including Indonesia, Laos, the Philippines, Thailand, and Timor-Leste [17]. Evidence suggests that this relationship is often mediated by other factors. Students who are physically attacked by their peers are more likely to be depressed, have difficulty winning friends, have poorer relationships with classmates, and experience loneliness [40].

The present study found that injury contributed to the increased likelihood of high psychological distress including suicidal behavior among young people. Injury may have poorly affected the physical and psychological health of adolescents, making them vulnerable to mental distress [41].

This study confirmed previous findings [42,43] showing an association between substance use, including current smoking and alcohol consumption, and suicide attempts in the adolescent population. An association between substance use and poor mental health or suicide attempts may refer to a clustering of risky behaviors.

Our findings showed that sexual intercourse was significantly associated with suicide attempts, which is in concordance with a previous research [23]. In that research, it was concluded that adolescents who reported having their first sexual intercourse before 14 years of age were more likely to have several psychological problems compared to adolescents who had their first sexual intercourse after the age of 14 years.

Recent studies have described that urban location [44] and feeling lonely [17] were strongly associated with suicide attempts, whereas our present study did not find an association between locations, loneliness, and suicide attempts.

Information on the risk factors of suicide attempts is fundamental for formulating an effective suicide-prevention program or intervention. Fostering socio-emotional life skills in adolescents is one of the four effective evidence-based interventions to prevent suicide, as stated in the LIVE LIFE implementation guideline [45]. In order to achieve that, school-based interventions in Mongolia should focus on strengthening general mental health, mitigating violence and bullying, and controlling and preventing risky behaviors, such as tobacco or alcohol use. Improved sexual health education, e.g., teaching strategies for refusing unwanted sex, might also contribute to reducing the risk of attempted suicide. Better self-esteem and development of life skills (including proper habits and lifestyle with good general and mental health, healthy eating behavior, and healthy decision-making) have been proven to decrease the risk of suicide among young people.

## 5. Limitations

The findings of this investigation must be viewed in light of its limitations. Firstly, the data were cross-sectional, so did not prove causation, but they did provide information on correlations. Secondly, a self-report questionnaire was applied, so it might be possible that the children provided invalid answers. Thirdly, this study focused on only in-school students meaning that the findings are not generalizable to all adolescents in the country. Fourthly, the involvement of parents/guardians may protect young people from risk factors, and the GSHS in Mongolia does not assess parental engagement. Finally, this questionnaire had no questions on sexual orientation, which could be the focus of further research.

## 6. Conclusions

High prevalence (32.1%) of suicide attempts was observed among school-attending adolescents in Mongolia, which remains a major public health problem. Several risk factors, including having no close friends, anxiety-induced sleep disturbance, frequent bullying, victimization, having been frequently physically attacked or injured, current tobacco use, alcohol drinking, and having had sexual intercourse were connected with students’ suicide attempts, whereas an increase in a student’s age by 1 year was a protective factor. Our results reinforce that the problem is increasing among school-going students, and the results may call the attention of the Mongolian government to the need to develop an independent and comprehensive adolescent health policy covering physical, mental, and behavioral problems of the adolescents in Mongolia. Findings of our study suggest that suicide-prevention programs should focus on encouraging general mental health. Important elements of that include mitigating violence and bullying, as well as controlling and preventing risky behaviors such as substance use (smoking and alcohol). Improving sexual health education, e.g., teaching strategies for refusing unwanted sex, might also contribute to a reduced risk for attempted suicides in the Mongolian adolescent population.

## Figures and Tables

**Table 1 ijerph-19-02991-t001:** Sample characteristics of school-attending adolescents included in the study by sex.

	Total (2850)		Male (1254)		Female (1596)	
	N	%	N	%	N	%
Suicide attempts						
Yes	916	32.1	417	33.3	499	31.3
No	1934	67.9	837	67.3	1097	68.7
Demographic factors						
Median age in year (IQR *)	15	−3	15	−2	15	−3
Place of residence						
Urban	916	32.1	412	32.9	504	31.6
Rural	1934	67.9	842	67.1	1092	68.4
Mental distress						
Close friend						
No	154	5.4	53	4.2	101	6.3
Yes	2683	94.1	1195	95.3	1488	93.2
Missing	13	0.5	6	0.5	7	0.4
Anxiety						
Yes	752	26.4	266	21.2	486	30.5
No	2094	73.5	987	78.7	1107	69.4
Missing	4	0.1	1	0.1	3	0.2
Feeling lonely						
Yes	1254	44	433	34.5	821	51.4
No	1594	55.9	819	65.3	775	48.6
Missing	2	0.1	2	0.2		
Injury and violence						
Being bullied						
Yes	1144	40.1	568	45.3	576	36.1
No	1699	59.6	682	54.4	1017	63.7
Missing	7	0.2	4	0.3	3	0.2
Being physically attacked						
Yes	1145	40.2	598	47.7	547	34.3
No	1703	59.8	655	52.2	1048	65.7
Missing	2	0.1	1	0.1	1	0.1
Injury						
Yes	1472	51.6	738	58.9	734	46
No	1375	48.2	514	41	861	53.9
Missing	3	0.1	2	0.2	1	0.1
Risky behaviors						
Cigarette smoking						
Yes	894	31.4	469	37.4	425	26.6
No	1950	68.4	780	62.2	1170	73.3
Missing	6	0.2	5	0.4	1	0.1
Alcohol drinking						
Yes	824	28.9	400	31.9	424	26.6
No	2025	71.1	854	68.1	1171	73.4
Missing	1	0	0	0	1	0.1
Sexual intercourse						
Yes	349	12.2	238	19	111	7
No	2491	87.4	1007	80.3	1484	93
Missing	10	0.4	9	0.7	1	0.1

* IQR: interquartile range. Missing: no answer was provided.

**Table 2 ijerph-19-02991-t002:** Results of univariable logistic regression analysis for suicide attempts by sex.

	Total			Male			Female		
	UAOR	95% CI	*p* Value	UAOR	95% CI	*p* Value	UAOR	95% CI	*p* Value
Demographic factors									
Age * (year)	0.87	0.83–0.91	<0.001	0.89	0.82–0.95	0.002	0.85	0.80–0.91	<0.001
Urban location	1.41	1.19–1.66	<0.001	1.15	0.90–1.48	0.251	1.65	1.32–2.07	<0.001
Mental distress									
No close friends	0.60	0.41–0.89	0.011	0.94	0.52–1.70	0.852	0.45	0.27–0.77	0.003
Anxiety	1.34	1.12–1.59	<0.001	1.19	0.90–1.58	0.214	1.47	1.18–1.85	0.001
Feeling lonely	1.17	1.00–1.38	0.042	1.11	0.87–1.43	0.369	1.26	1.02–1.56	0.028
Injury and violence									
Being bullied	31.93	25.42–40.11	<0.001	32.03	22.19–46.23	<0.001	34.41	25.52–46.40	<0.001
Physically attacked	33.03	26.27–41.53	<0.001	30.18	20.74–43.92	<0.001	42.57	31.34–57.81	<0.001
Injury	21.27	16.60–27.27	<0.001	20.21	13.36–30.56	<0.001	23.60	17.23–32.34	<0.001
Risky behaviors									
Smoking	69.89	54.47–89.67	<0.001	43.02	30.39–60.89	<0.001	217.69	133.85–354.01	<0.001
Alcohol drinking	145.73	107.45–197.65	<0.001	125.84	82.43–192.10	<0.001	190.89	119.50–304.93	<0.001
Sexual intercourse	1.22	0.96–1.54	0.092	1.14	0.84–1.53	0.386	1.31	0.88–1.96	0.179

* Age: continuous variable. UAOR: unadjusted odds ratio. 95% CI: 95% confidence interval.

**Table 3 ijerph-19-02991-t003:** Results of multivariable stepwise logistic regression analysis for suicide attempts by sex.

	Total			Male			Female		
	* AOR	95% CI	*p* Value	AOR	95% CI	*p* Value	AOR	95% CI	*p* Value
Demographic factors									
Age (year)	0.89	0.80–0.99	0.036	-	-	-	0.84	0.73–0.97	0.017
Mental distress									
No close friends	3.3	2.02–5.40	<0.001	5.68	2.55–12.63	<0.001	2.26	1.19–4.28	0.012
Anxiety	1.51	1.09–2.10	0.013	-	-	-	2.02	1.30–3.12	0.002
Injury and violence									
Being bullied	2.41	1.67–3.47	<0.001	2.88	1.65–4.99	<0.001	2.22	1.35–3.65	0.001
Physically attacked	2.59	1.80–3.73	<0.001	3.25	1.84–5.74	<0.001	2.76	1.67–4.54	<0.001
Injured	2.35	1.66–3.31	<0.001	2.78	1.59–4.87	<0.001	2.29	1.46–3.59	<0.001
Risky behaviors									
Cigarette smoking	5.02	3.17–7.95	<0.001	3.93	2.19–7.07	<0.001	13.62	5.55–33.45	<0.001
Alcohol drinking	12.83	8.00–20.58	<0.001	17.58	9.59–32.24	<0.001	5.17	2.07–12.90	<0.001
Sexual intercourse	1.86	1.17–2.95	0.008	2.14	1.16–3.95	0.014	-	-	-
Nagelkerke R^2^		0.779			0.793			0.781	

Reference categories: rural location, had close friends, did not report anxiety, did not report loneliness, did not report being physically attacked, did not suffer any injury, did not report being bullied, no cigarette smoking, no alcohol drinking, and no sexual intercourse. * AOR: adjusted odds ratio. Variables not entered in the models: Total: living place and feeling lonely; male: age, living place, anxiety, and feeling lonely; female: living place, feeling lonely, and sexual intercourse.

## Data Availability

Datasets generated during the study can be obtained directly from the third author at suvd552001@gmail.com.

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
