# Peer review of "Suicide Attempts among School-Attending Adolescents in Mongolia: Associated Factors and Gender Differences"

_ijerph, 2022, doi:10.3390/ijerph19052991_

Round 1

Reviewer 1 Report

The Title is appropriate and reflects the contents of the article.

The Abstract is focused on data and adequately summarizes the information found in the body of the article.

The body of the article adheres to the journal guidelines: the Introduction places the study in a broad context and defines the purpose of the work and its significance, the Materials and Methods are described with sufficient detail, the Results provide a concise and precise description of the experimental results, their interpretation as well as the experimental conclusions that can be drawn, the Discussion elaborates on the results and how they can be interpreted in perspective of existing research, the Conclusion is supported by data. The included tables are in accordance with journal guidelines.

With regards to its contents, the article has an adequate degree of novelty and practical significance, with sufficient sample size, accurate methods and appropriate statistical tests, that render the study reproducible

English spelling and general phrasing require improving upon. In this regard, please consider the following annotations:

Line 16, please write "year" instead of "years".

Line 18 and throughout the text,  please put the appropriate article before "suicide attempt".

Line 20 and line 171, please write "less likely to have close friends" instead of "more likely to have no close friend".

Line 21, please put the appropriate qualitative details concerning the variable " injury and violence".

Line 58, please put the appropriate article before "close friend".

Line 66-67, please clarify the phrasing " they tend to have suicide attempt than males" according to the information provided by the reference.

Line 145, please put "were" instead of "was".

Line 152 and line 155, please put "cigarettes" instead of "cigarette".

Line 166, please rephrase "each one year's increase of age was associated with less likely suicide attempt" in a more streamlined manner. An example would be "each one year increase in age was associated with progressively fewer suicide attempts".

Line 170, please insert a full stop before "Altogether".

Line 171, please insert the appropriate qualitative details concerning the variable "being bullied", as well as correct tense.

Line 176, please rephrase the following enumeration "no anxiety, no lonely, no physically attacked, no injury, no being bullied" in a more adequate English manner. An example would be: "did not report/ exhibit anxiety, did not report being physically attacked, did not suffer any injury, did not report being bullied".

Lines 186, 187, 190, 194, 197, 200, 204, 208, 213, 224, 225 and 234 as well as Title, please use "suicide attempts" instead of "suicide attempt".

Line 217, please rephrase the sentence "have greater struggle making friends" in a more adequate English manner.

Line 221, please check the spelling of the word "psychical" and replace it with the correct term

Line 227, please use "show" instead of "showing" and rephrase the sentence in a streamlined manner.

Line 230, please remove "at" from " at after 14 years old".

Line 257, please insert a comma after "bullying".

Author Response

We are thankful for your helpful comments, which have helped us to improve our manuscript. We hope that we have addressed your comments adequately.

The English spelling of the paper has been revised by an English language expert.

Comments and Answers

  1. Line 16, please write "year" instead of "years".

Answer: It has been revised.

  1. Line 18 and throughout the text please put the appropriate article before "suicide attempt".

Answer: It has been changed throughout the paper.

  1. Line 20 and line 171, please write "less likely to have close friends" instead of "more likely to have no close friend".

Answer: It has been revised.

  1. Line 21, please put the appropriate qualitative details concerning the variable " injury and violence".

Answer: It has been revised: “Male suicide attempters were less likely to have close friends and more likely to have injuries, being physically attacked, being bullied, more likely to smoke, drink alcohol, and have had sexual intercourse.”

A more detailed description about “injury and violence” related questions have been added to Methods according to the description provided in the questionnaire:

Bullying occurs when one or more students or someone else about your age teases, threatens, ignores, spreads rumors about, hits, shoves, or hurts another person over and over again.

An injury is serious when it makes you miss at least one full day of usual activities (such as, school, sports, or a job) or requires treatment by a doctor or nurse.

A physical attack occurs when one or more people hit or strike someone, or when one or more people hurt another person with a weapon (such as, a stick, knife, or gun).

  1. Line 58, please put the appropriate article before "close friend".

Answer: It has been changed throughout the paper.

  1. Line 66-67, please clarify the phrasing " they tend to have suicide attempt than males" according to the information provided by the reference.

Answer: It has been revised as “…females tend to have more suicide attempts than males”.

  1. Line 145, please put "were" instead of "was".

Answer: It has been revised as “were”.

  1. Line 152 and line 155, please put "cigarettes" instead of "cigarette".

Answer: It has been revised as “cigarettes”.

  1. Line 166, please rephrase "each one year's increase of age was associated with less likely suicide attempt" in a more streamlined manner. An example would be "each one year increase in age was associated with progressively fewer suicide attempts".

Answer: It has been revised: “Age remained significant in case of females; each one year increase in age was associated with progressively fewer suicide attempts (AOR: 0.84).”

  1. Line 170, please insert a full stop before "Altogether".

Answer: It has been revised.

  1. Line 171, please insert the appropriate qualitative details concerning the variable "being bullied", as well as correct tense.

Answer: It has been revised: “Altogether, male suicide attempters were less likely to have close friends, and more likely to having been bullied, physically attacked, injured, smoke cigarettes, and drink alcohol, and have had a sexual intercourse.”

  1. Line 176, please rephrase the following enumeration "no anxiety, no lonely, no physically attacked, no injury, no being bullied" in a more adequate English manner. An example would be: "did not report/ exhibit anxiety, did not report being physically attacked, did not suffer any injury, did not report being bullied".

Answer: It has been revised: “had close friends, did not report anxiety, did not report loneliness, did not report being physically attacked, did not suffer any injury, did not report being bullied, no cigarette smoking, no alcohol drinking, and no sexual intercourse.”

  1. Lines 186, 187, 190, 194, 197, 200, 204, 208, 213, 224, 225 and 234 as well as Title, please use "suicide attempts" instead of "suicide attempt".

Answer: It has been revised as “suicide attempts” throughout the paper.

  1. Line 217, please rephrase the sentence "have greater struggle making friends" in a more adequate English manner.

Answer: It has been revised: “Students who are physically attacked by their peers are more likely to be depressed, have difficulty to win friends, have poorer relationships with classmates, and experience loneliness.”

  1. Line 221, please check the spelling of the word "psychical" and replace it with the correct term

Answer: It has been revised as “psychological”.

  1. Line 227, please use "show" instead of "showing" and rephrase the sentence in a streamlined manner.

Answer: It has been revised as “show”.

  1. Line 230, please remove "at" from " at after 14 years old".

Answer: It has been removed.

  1. Line 257, please insert a comma after "bullying".

Answer: It has been inserted.

Reviewer 2 Report

I commend the authors for describing this critical and timely issue. The paper is interesting and well-written; however, I would like to highlight some issues that merit revision:

On the basis of the questionnaire presented there are many interesting data, and in this regard, I ask the authors if it has been evaluated, within the question about bullying, the presence of discriminatory acts on the basis of sexual orientation, and also, as currently present in the DSM-5, I ask if it has been detected the presence of any disorders of gender identity, a factor not infrequently involved in self-harming thoughts or real suicidal attitudes. Please, if necessary, briefly describe results found in this sense or add this missing data between the limitations.

Author Response

We are thankful for your helpful comments, which have helped us to improve our manuscript. We hope that we have addressed your comments adequately.

Comments and Answers

I commend the authors for describing this critical and timely issue. The paper is interesting and well-written; however, I would like to highlight some issues that merit revision:

On the basis of the questionnaire presented there are many interesting data, and in this regard, I ask the authors if it has been evaluated, within the question about bullying, the presence of discriminatory acts on the basis of sexual orientation, and also, as currently present in the DSM-5, I ask if it has been detected the presence of any disorders of gender identity, a factor not infrequently involved in self-harming thoughts or real suicidal attitudes. Please, if necessary, briefly describe results found in this sense or add this missing data between the limitations.

Answer:

The questionnaire was developed by the WHO and the CDC, which is fixed format. It did not cover any issue related to sexual orientation.

Global School-based Health Survey (GSHS) is a cross-sectional study, which does not prove causation, only provide information on correlation of health risk behaviors among school-going adolescents. We have included this information in the limitation section.

Considering your suggestion, this sentence has been involved into the limitations: “Finally, this questionnaire had no questions on sexual orientation, which could be the issue of a further research.”

Reviewer 3 Report

This is a tragic and interesting paper  The resounding conclusion is that  in 2019 approximately1/3 of Mongolian adolescents reported they had attempted suicide in the past 12 months.     This is astounding.   There are a few suggestions I think can make the paper better.   First, this is a self report questionnaire.  If there is any evidence from ERs that the rate is this high, it should be added.  If there is no data, that should also be stated.  Second, the authors note this rate is much higher than the rate of 10% in 2013, which also is astronomically high.   They should offer some comments on what has changed in these 6 years--has there been environmental degradation, has the poverty rate changed, is mining taking hold as a new industry and changing mores.  Even if one can not determine which factor is important for the increase it would be good to suggest anything, especially since these authors are in Ulann baatar, Mongolia, and understand the situation on the ground.  

Author Response

We are thankful for your helpful comments, which have helped us to improve our manuscript. We hope that we have addressed your comments adequately.

Comments and Suggestions for Authors

This is a tragic and interesting paper. The resounding conclusion is that in 2019 approximately1/3 of Mongolian adolescents reported they had attempted suicide in the past 12 months.     This is astounding.   There are a few suggestions I think can make the paper better.   

  1. First, this is a self-report questionnaire.  If there is any evidence from ERs that the rate is this high, it should be added.  If there is no data, that should also be stated. 

Answer:

We cannot provide evidence supporting that the rate is so high. According to a summary from the UNICEF, Mongolia has five times as high rate of average adolescent suicide as in East Asia and the Pacific region, and the mortality is increasing (Lines 40-43, Reference 5 in the paper). We think that the self-reported suicide attempt rate is correlated with this trend.

So, it has been involved into the limitations section: “Secondly, a self-report questionnaire was applied, so it might be possible that the children provided invalid answers.”

  1. Second, the authors note this rate is much higher than the rate of 10% in 2013, which also is astronomically high.   They should offer some comments on what has changed in these 6 years--has there been environmental degradation, has the poverty rate changed, is mining taking hold as a new industry and changing mores.  Even if one cannot determine which factor is important for the increase it would be good to suggest anything, especially since these authors are in Ulaanbaatar, Mongolia, and understand the situation on the ground. 

Answer:

This sentence has been added to the Introduction (Line 70-72):

“Although, findings of studies using Mongolian GSHS 2010 and 2013 revealed that suicide attempts had increased significantly, and no suicide prevention program has been implemented in Mongolia in the previous decade. Considering it, the aim…”

This sentence has been added to the Conclusion:

“Our results reinforce that the problem is increasing among school-going students, and the results may call the attention of the Mongolian government for the need to develop an independent and comprehensive adolescent health policy covering physical, mental, and behavioral problems of the adolescents in Mongolia. Findings of our study….”
